# 12-year observation of tweets about rubella in Japan: A retrospective infodemiology study

**Yukie Sano**[1]*, **Ai Hori**[2]

**1** Faculty of Engineering, Information and Systems, University of Tsukuba, Tsukuba, Ibaraki, Japan, **2** Faculty of Medicine, University of Tsukuba, Tsukuba, Ibaraki, Japan

* sano@sk.tsukuba.ac.jp

## Abstract

Although rubella is an infectious disease that can be prevented by vaccination, there have been periodic epidemics in Japan, mainly among adult males. One of the reasons for this is the lack of interest in vaccination among the target adult male population. To clarify the reality of the discussion about rubella and provide basic resource for enlightening activities for rubella prevention, we collected and analyzed Twitter posts about rubella in Japanese between January 2010 and May 2022. We examined time series, number of tweets per account, tweeted contents, and retweet network. We found that the weekly number of rubella reports and the number of Twitter posts fluctuate simultaneously. During the 2018 rubella epidemic, the number of tweets increased due to the start of the rubella routine vaccination program and the use of cartoons to raise awareness. While 80% of the accounts posted three times or fewer during the period, some accounts posted multiple times per day for more than 12 years. Medical terms such as vaccines and antibodies were frequently used in the tweet contexts. In the retweet activity, a variety of actors, including mass media, medical professionals, and even rubella sufferers, contributed to disseminate rubella-related information.

## Introduction

### Rubella in Japan

Rubella is a vaccine-preventable disease; rubella-containing vaccine and measles-rubella vaccine are approved in Japan as of 2023. However, periodic outbreaks have occurred in Japan, mainly among adult men who did not have the opportunity to be vaccinated during their childhood. While women of the same age had the opportunity to be vaccinated, men born before April 1, 1979 did not have the opportunity to be vaccinated even once [1].

In the 2012–2013 rubella epidemic, there were more than 14,000 reported cases of infection [2], and 45 cases of congenital rubella syndrome were reported [3] with the key clinical characteristic of conjunctivitis [4] After that, the Japanese government set a goal of eliminating rubella by 2020 [5]. The municipal government offered partial expense support for rubella antibody testing and vaccination [6]. However, the voluntary rubella vaccination rate

**Data Availability Statement:** Due to Twitter's terms of service, we cannot share the raw data used in this study. However, we share the daily/ weekly number of tweets, the number of tweets per

account, and frequently appeared words in the supplement for replication purposes.

**Funding:** Y.S. received the Japan Society for the Promotion of Science KAKENHI Grant Number 20K19928. A.H. received the Japan Society for the Promotion of Science KAKENHI Grant Number 20K10467 and Health Labour Sciences Research Grant number 21HA2014. The funders had no role in study design, data collection and analysis, decision to publish, or preparation of the manuscript.

**Competing interests:** The authors have declared that no competing interests exist.

remained low, especially among men who did not plan to have babies [7]. A catch-up vaccination program for adult men was not developed at that time.

Then the rubella outbreak occurred again in 2018. The U.S. Centers for Disease Control and Prevention (CDC) recommended that unvaccinated people and pregnant women refrain from traveling to Japan [8]. Due to concerns about the negative impact of the rubella outbreak on the upcoming 2020 Rugby World Cup and Tokyo Olympic/Paralympic Games [9], the rubella routine vaccination program was initiated in 2019. In the new rubella routine vaccination program, free vaccination was offered to men born between fiscal years 1962 and 1978 who were preliminarily tested for antibodies freely and found to have low antibody titer [10]. The promotion measures were introduced to allow working-age men to take rubella antibody tests on the occasion of annual health checkups at their workplaces or in their communities [11]. However, use of the catch-up program remained low at 20% [12] of the eligible population at the end of 2021, and there was a risk of a resurgence of rubella outbreaks under the current circumstances. Therefore, the Japanese government decided to extend the rubella routine vaccination program for three years from 2022 [13], and further promotion of the program is required.

One of the reasons for the stagnation of routine rubella vaccination is the lack of interest among the target population, men in their 40s and 50s. Compared to women, adult men are known to be less likely to take actions such as checking rubella vaccination history, antibody testing, and immunization [14]. Factors associated with the use of the routine immunization program include knowing the government's recommendation, having acquaintances who have used the program, and being aware that their generation did not have the opportunity to be immunized [15].

## Infection disease and SNS

Monitoring and enlightining of infectious diseases using SNS such as Facebook and Twitter has been active in public health targeting measles [16–23], human papillomavirus (HPV) [24–26], Ebola [27, 28] and zika [29, 30] even before the COVID-19 infodemic [31–34]. During the rubella epidemic in the Netherlands in 2013, the number of rubella cases and online news increased simultaneously, leading to an increase in tweets (Twitter posts) [16]. It is also reported that tweets increase in response to a main political event such as mandatory immunizations in Italy [21]. Public agencies such as the CDC and World Health Organization (WHO) were influential in disseminating information during the zika outbreak in 2015–2016 [29].

These previous studies have mainly analyzed a couple of years of epidemics. On the contrary, few studies exist that track the topic of specific infectious diseases, especially rubella, over ten years or more, including periods of no epidemic.

According to a 2021 survey, the percentage of individuals using social networking services (SNS) in Japan reached 78.7% [35]. In order to arouse widespread interest in rubella, the Ministry of Health, Labor and Welfare (MHLW) in Japan has begun actively using Twitter, one of the popular SNS in Japan, to announce rubella-related information. Similarly, rubella-related actors, such as sufferers or family members, and doctors are also actively using Twitter to disseminate information.

## Purpose of the study

The purpose of this study is to clarify the reality of the discussion about rubella on Japanese Twitter space and provide basic data for enlightening activities on rubella prevention. We analyzed posts on SNS for rubella over a period of 12 years and 5 months. In particular, we analyzed posts on Twitter, which is said to have 45 million users in Japan as of 2017 [36]. There

has been little debate in favor of or against the rubella vaccine in Japan, but the lack of interest in rubella has been problematic. Therefore, it is crucial to observe how this pubic interest in rubella has evolved.

In the following sections, we first provide an overview of the data and the main analytical methods used. Next, we present the results of our analysis of long-term fluctuations and the content of the posts. Finally, we summarize the results of this analysis and discuss the significance of the study.

## Methodology

### Data

In this study, Twitter Academic API was used to retrieve tweets about rubella written in Japanese. Our project was approved by Twitter's Academic Research Program, which gave us access to the full archive data (https://developer.twitter.com/en/use-cases/do-research/academic-research). The acquisition condition was defined as tweets containing "rubella" expressed in either Kanji (Chinese characters) or Hiragana from January 1, 2010 to May 31, 2022. We collected these rubella-related tweets posted publicly on Twitter.

Usually, when text-based tweets are collected, they may contain spam tweets intended as advertisements. In the case of rubella in Japanese, there were almost no spam-related tweets, probably because "rubella" is not a very common word. Therefore, no special filtering process was used in this data collection.

We separated an original tweet and a retweet based on whether a tweet included "RT" at the beginning of the body of the tweet. Therefore, quoted retweets that precede the quoted text with their own opinion were treated as original tweets.

### Overview of analysis

This study aimed to provide a basic analysis of rubella-related Twitter posts; therefore, we focus on the following four points.

- Time series of rubella-related tweets and rubella reports

- Number of tweets per account

- Visualization of content

- Retweet network

**Time series of tweets and reports.** Weekly counts of tweets were conducted to determine the overall fluctuation of tweets about rubella and compare them with the number of rubella reports [2]. The number of rubella reports was counted by downloading the information available on the website (https://www.niid.go.jp/niid/ja/hassei/3086-rubella-sokuhou-rireki.html). To be consistent with the time period for the reported cases, we used the tweet time period of ten years from 2012. We then operated `StatsLinearCorrelationTest` by Igor Pro 9.0 (WaveMetrics, Inc.) to calculate Pearson's correlation coefficient.

**Number of tweets per account.** To determine the extent to which accounts posting about rubella, we counted the number of tweets per account throughout the entire period from 2010 to 2022. In addition to MHLW, the top two accounts with the most tweets were individually web-surveyed to determine whom they were inferred to be based on publicly available information. In case that these accounts agreed, we included their screen names in the result.

**Visualization of content.** In order to understand tweet content, we visualized tweets using wordcloud (http://amueller.github.io/word_cloud/). We created a wordcloud using the top 150 most frequent words for ease of visualization. Here, as a pre-processing of the tweet body, the words were segmented using NEologd [37], a dictionary that also supports Internet slang, omitting punctuation and symbols.

We automatically translated the original Japanese words into English using Google translate (https://translate.google.com/). Then, the author (AH) with domain knowledge of medical terminology checked the translation and manually corrected. At that time, we ensured that one Japanese word corresponds to one English word and is not duplicated.

**Retweet network.** We built a retweet network that a node represents each account and a link represents retweet between them. If there were multiple retweets between the same accounts, we reflected as a link weight.

For the network visualization, we showed the retweet network by extracting only those link weights greater than 100. For the top fourteen accounts with their link weights, we individually web-surveyed to determine whom they were inferred to be based on publicly available information including number of followers. For those among top fourteen link weight accounts that they agreed, their screen name was stipulated in the results. The size of the nodes was proportional to the size of the followers after taking the logarithm. The color and size of the nodes were divided between mass media/government agencies, doctors, and rubella-related actors.

## Results

We collected 2,410,868 tweets from 575,311 accounts in total. The percentage of retweets was 64.9%. The average number of tweets (sum of both original tweets and retweets) per day for the entire period covered was 532 (187 original tweets and 345 retweets). Thus, daily rubella-related tweets were very small compared to the Japanese Twitter population.

### Time series of tweets and reports

Fig 1 shows the weekly number of rubella reports and the rubella-related tweets. The changes in original tweets and retweets were almost synchronous, with a correlation coefficient $r = 0.71$ ($p < .001$, 95%CI 0.67–0.75). The overall changes were also similar to the number of rubella reports, with an increase in the number of tweets in 2013 and 2018. In particular, the number of reports and the number of original tweets fluctuate simultaneously, with a correlation coefficient $r = 0.69$ ($p < .001$, 95%CI 0.65–0.74). On the contrary, the correlation coefficient between the number of reports and the number of retweets was less similar, $r = 0.21$ ($p < .001$, 95%CI 0.13–0.29).

On a daily basis, the day with the largest number of original tweets was December 11, 2018 with 4,820 tweets. Fig 1 shows weekly numbers that have the highest peak in the middle of 2018 for original tweets. The day with the most retweets was October 11, 2018 with 27,068 retweets. The median number of daily tweets increased more than double before and after 2018, when the second epidemic occurred; the value varied from 153 (95 original tweets and 57 retweets) in 2017 to 555 (219 original tweets and 333 retweets) in 2019.

### Number of tweets per account

Approximately 320 thousand accounts, or 56.6% of the total accounts, made only one tweet, 90 thousand accounts made two tweets, followed by 40 thousand accounts with three tweets (Fig 2(a) and S1 Appendix for more details).

Contrarily, one account made 33,073 posts during this period. This account (@Dr_Rasu-Karu) is a doctor with about ten thousand followers, and has been regularly tweeting about

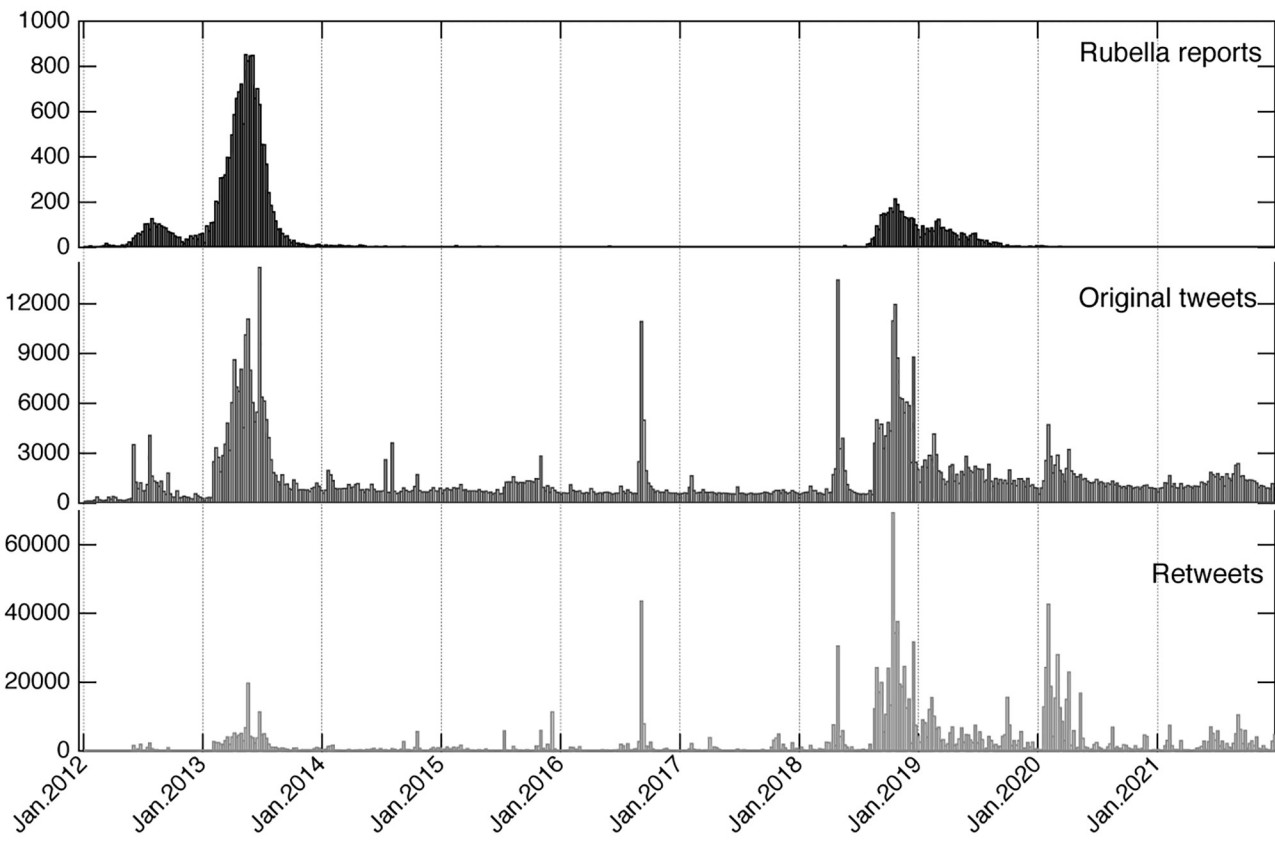

**Fig 1. Weekly time series of tweets and rubella reports.** From the top row, the weekly number of rubella reports, original tweets, and retweets, respectively.

rubella, using a bot that automatically posts tweets. The next highest account (@knimama) is a family member of those affected by congenital rubella syndrome who made 19,565 tweets. This account was very active in the rubella eradication effort, and although it does not use any automated framework, it made numerous posts. In addition, the official account of MHLW (@MHLWitter) was also active in tweeting information on rubella, posting 1,136 tweets during the period. MHLW was the 56th most tweeted among the 575,311 accounts. Thus, overall, the number of tweets about rubella per account was highly skewed and showed power-law distribution $P(x) \sim x^{-\alpha}$ with its exponent $\alpha$ estimated [38] as 1.8 (Fig 2(b)).

## Visualization of written content

Fig 3 shows wordclouds created with original tweets and retweets. The most tweeted words were "Vaccination" for original tweets and "Vaccine" for retweets. Although these words have very similar meanings, their Japanese translations differ. "Vaccination" is a four-character Kanji word that refers to the act of inoculation, while "Vaccine" refers to the vaccine itself (S1 Appendix).

Fig 4 shows wordclouds of original tweets on December 10, 2018 (the day with the most original tweets) and retweets on October 11, 2018 (the day with the most retweets). The most tweeted word on December 10, 2018 was the same as the overall trend:"Vaccination." The

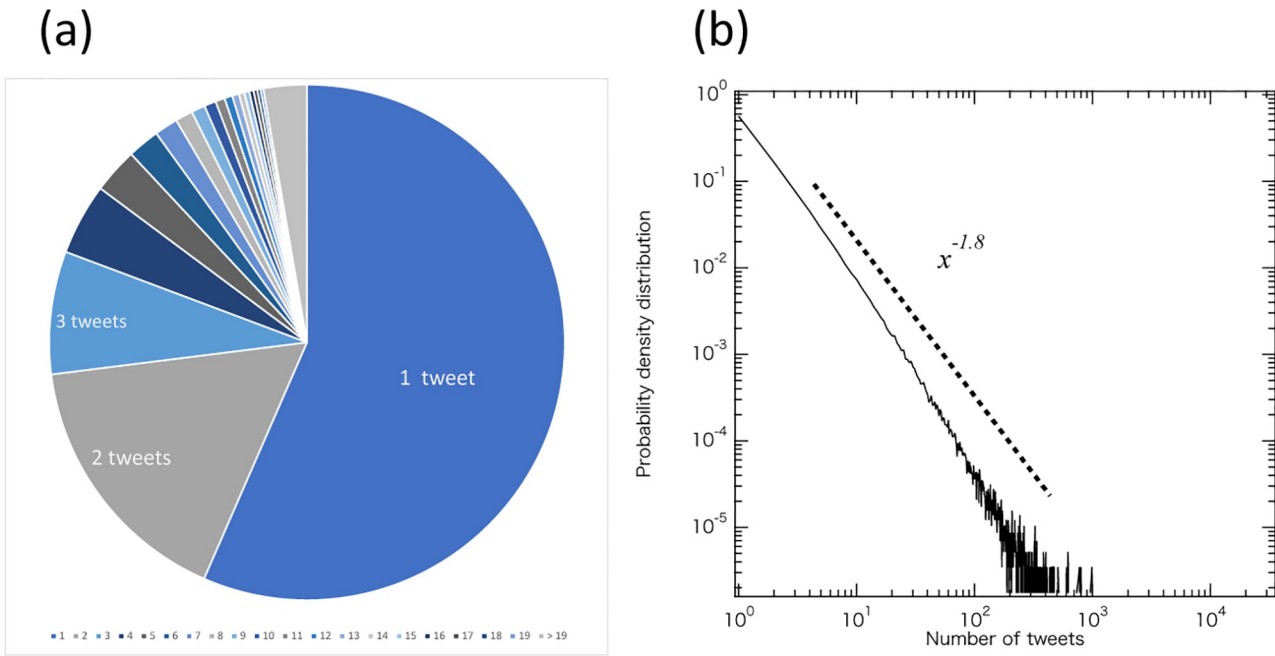

**Fig 2. Number of tweets per account.** (a) Ratio of tweets per account. The majority of the tweets were made by only a small number of users. (b) Distribution of number of tweets per account (logarithmic on both axes). The typical power distribution with an exponent of 1.8.

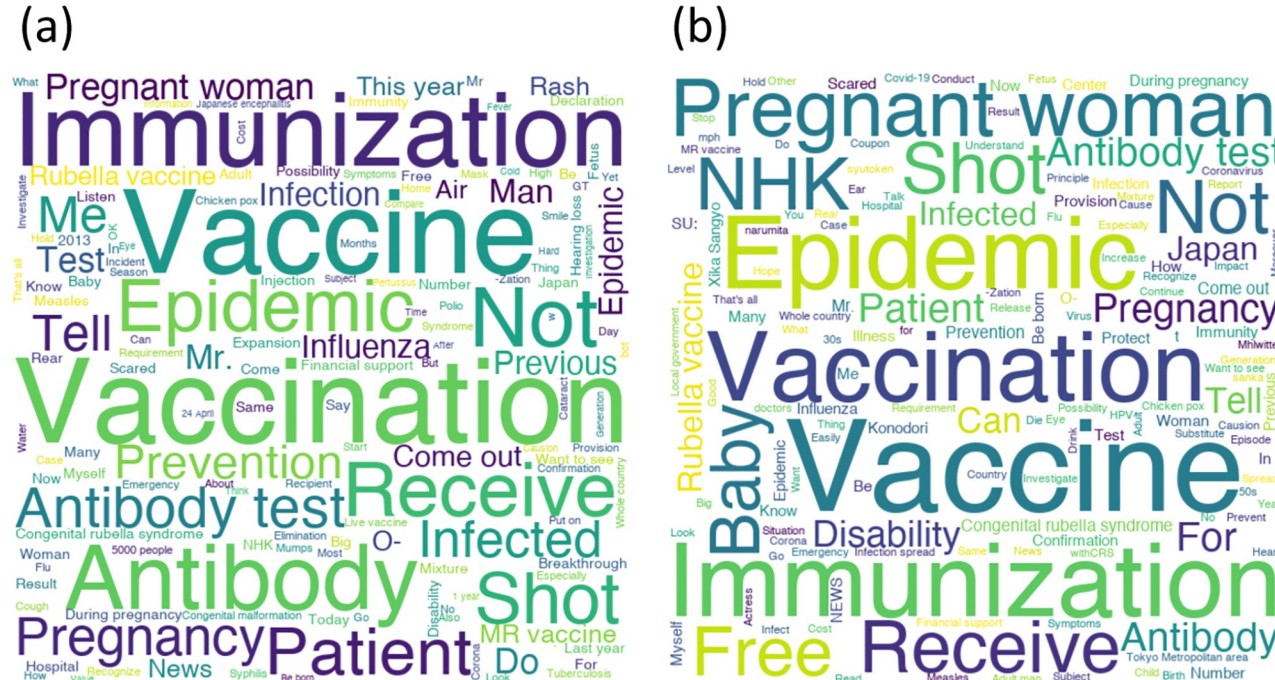

**Fig 3. Wordcloud created with the top 150 most frequently occurring words during the entire period.** (a): Original tweets only. (b): Retweets only.

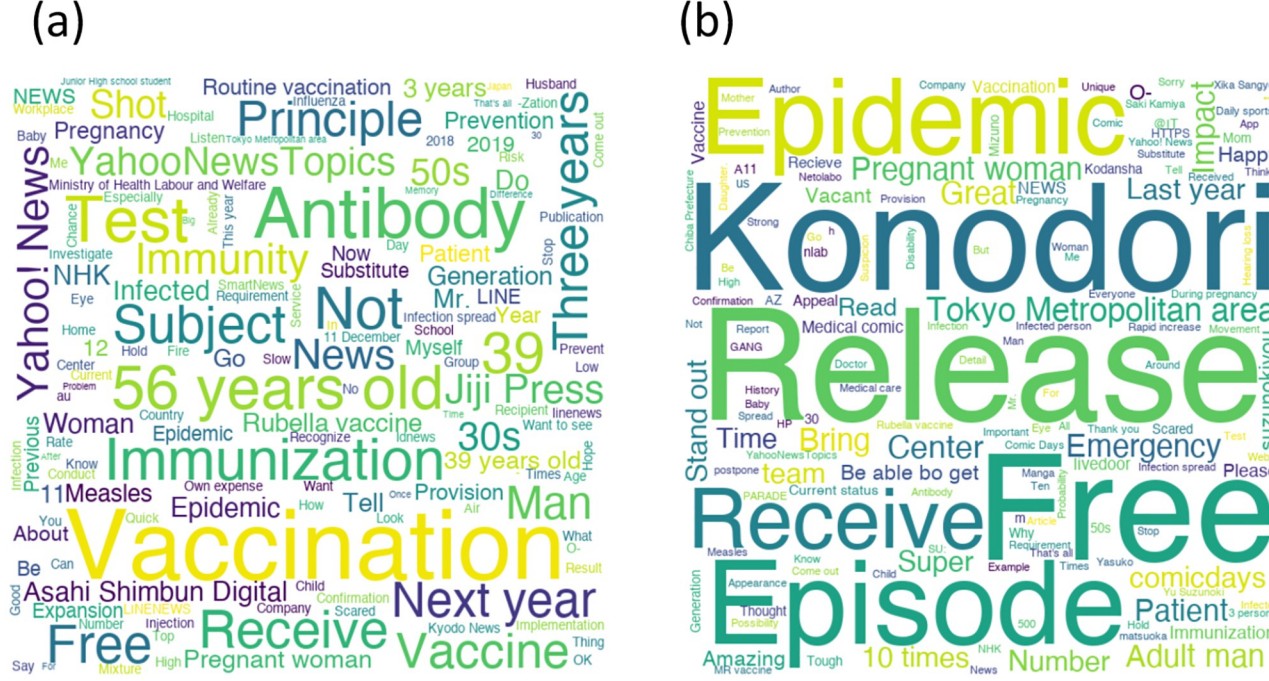

**Fig 4. Wordcloud created with the top 150 most frequently occurring words at the specific day.** (a): Original tweets only (Dec. 12th, 2018). (b): Retweets only (Oct. 11th, 2018).

most retweeted word on October 11, 2018 was "Release." The top 200 most frequent words with original Japanese forms appear in S1 Appendix.

### Retweet network

Fig 5 shows the retweet network of rubella-related tweets. In the network, top fourteen nodes with the largest number of links were divided into three types: the mass media (@nhk _news, @nhk_seikatu) and MHLW (@MHLWitter), medical professionals (@narumita, @Dr_Rasu-Karu, @syutoken_sanka, @imamura_kansen), and rubella-related actors (@knimama, @stop_-fuushin). Within this network, the nodes with the highest degree centrality, which is an index proportional to the number of links, was the mass media account (@nhk_seikatu) with a large number of followers. Contrarily, the nodes with the highest betweeness centrality, which measures mediating shortest path [39], was the account of the rubella-related actor (@knimama).

## Discussion

Even rubella-related tweets are weak signals; we found that they have increased in recent years, particularly in response to news and related events. These results are consistent with SNS analyses of measles and other diseases conducted in other countries [19, 21].

### Time series of tweets and reports

The weekly number of rubella cases and corresponding tweets exhibit a significant correlation, particularly for original tweets ($r = 0.69$). Similar correlations between tweets and disease surveillance are reported in other studies. For instance, in the US, a study on influenza revealed a strong correlation ($r = 0.93$) between the number of tweets classified as valid and the rates of

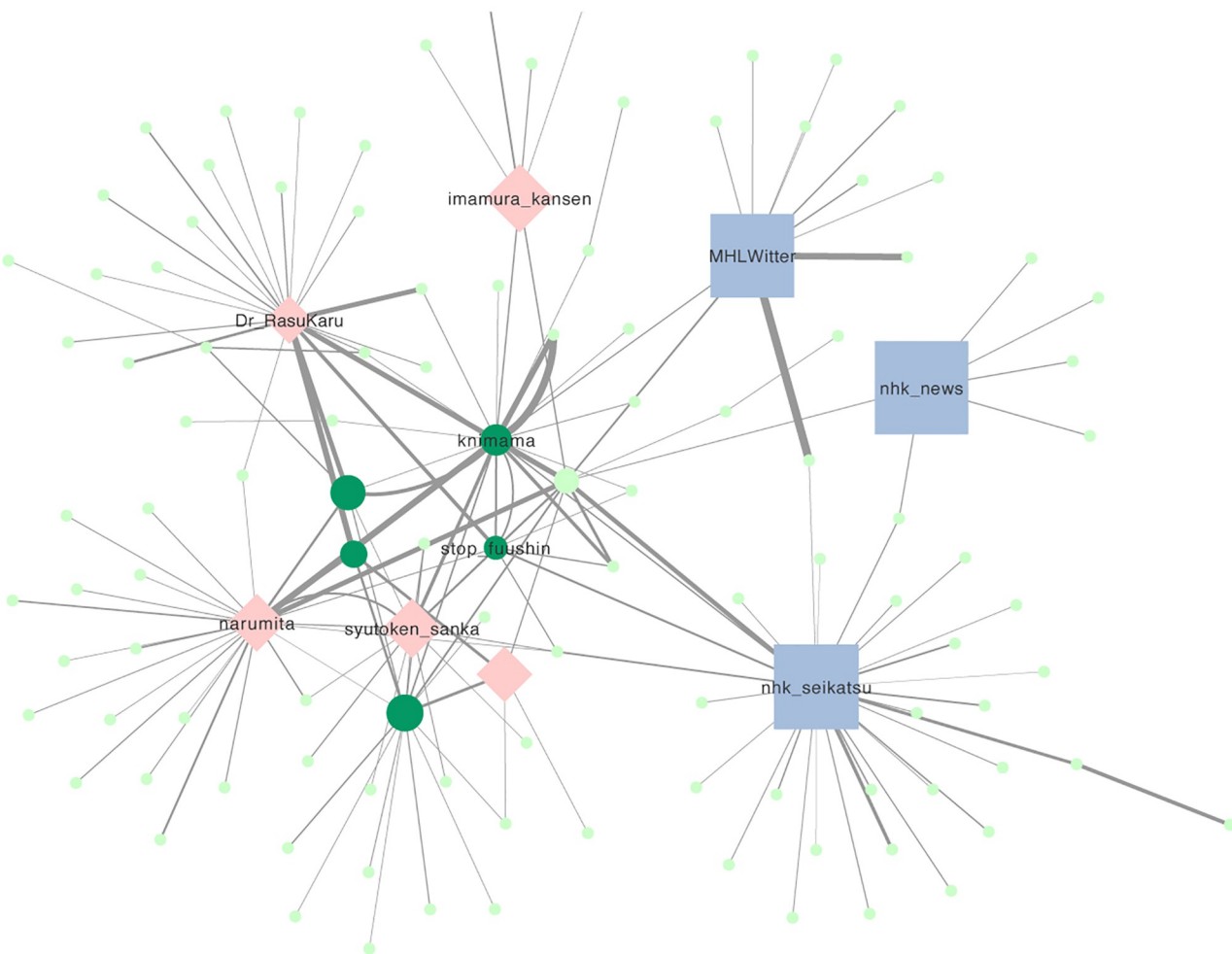

**Fig 5. Retweet network of rubella-related tweets.** Node size proportional to the logarithm number of followers as of February 2023. The blue squares represent the mass media and MHLW, the pink diamonds represent medical professionals, and the green circles represent the rubella-related actors.

influenza-like illness, although the strength of the relationship varied across different cities [40]. In the case of the 2013 measles outbreak in the Netherlands, the correlation between tweets and disease incidence was reported to be $r = 0.56$ [16]. Not only infectious diseases but also hay fever has been reported correlation ($r = 0.97$) in the UK [41]. In Japan's case, tweets related to allergic rhinitis were highly correlated ($r = 0.84$) with their drug sales [42]. We reached similar results using longer-term data.

The day with the largest number of original tweets was the day MHLW announced that vaccinations would be free for three years for men aged 39 to 56 as a measure against rubella. The day with the most retweets was the day the episode about rubella in the cartoon "Konodori" was released for free. "Konodori" is a popular medical cartoon featuring an obstetrician. On the day of the event, the cartoon created many tweets because it was published online for free to stimulate interest in rubella.

## Number of tweets per account

A small number of accounts continue to disseminate information on rubella actively for more than ten years. Although the Japanese MHLW was in the top 1% of most tweeted account in

our data, it was hardly the most influential in the Japanese Twitter space for rubella. Instead, individuals such as a doctor and a family member with congenital rubella syndrome were most active. A previous study also reports existence of hidden influential users who have a small number of followers but being retweeted multiple times as well as normal influential users who have a large number of followers in Ebola tweets [27]. We obtained similar results.

The majority of accounts that posted about rubella tweeted only once or twice. This suggests that the majority of accounts are interested in something other than rubella on a daily basis. These results suggest that Twitter's activities to raise awareness using tie-up projects about rubella may reach people who were not interested in the disease in some sense.

## Visualization of content

As an overall trend, many of the words that frequently appeared in both original tweets and retweets were common. Those common words were medical terms such as "Vaccination," "Vaccine," and "Antibody." There were also many tweets containing "Pregnant woman." This suggests that the majority of the tweets probably conveyed correct medical information that pregnant women infected with rubella may have children with congenital rubella syndrome, which can be prevented by vaccination.

On December 11, 2018, MHLW announced that antibody testing and vaccination would be free for men aged 39 to 56. At the time, major newspapers reported the news, and many people shared it on the web. This event caused "Man" and "Free" to appear on the day's wordclud, in addition to "Vaccination." The highest number of retweets occurred on October 11, 2018, with several characteristic words ("Release," "Konodori," "Free") appearing that were not present during the other periods. The event caused the episode about rubella in the cartoon "Konodori" released for free to arouse public interest in rubella. It is suggested that press releases using tie-up projects with popular content can be effective in promoting the spread of public health information.

The word "Free" was shared on both days, with the largest number of original tweets and retweets, even though it appeared in separate contexts. These were free vaccinations in the case of the original tweet and free cartoon publication in the case of the retweet. The word "Free" may be attractive and have attracted people, and it is important to examine this finding for future health-promoting campaigns.

## Retweet network

The retweet network revealed the existence of important accounts of dissemination of rubella information in Japanese Twitter space. Consistent with the previous studies [18, 27, 29], tweets from the mass media and MHLW, which have a large number of followers, were frequently retweeted. On the contrary, although the numbers of followers were smaller, accounts of individual doctors and rubella-related actors also frequently retweeted or were being retweeted. Zika's research on Twitter also pointed out the existence of a small number of hidden influencers who have a small number of followers but a large number of retweets [27]. Our result also confirmed a similar phenomenon.

The retweet network revealed a cooperative effort between mass media with a large number of followers and individuals with a small number of followers who mediate the dissemination of information. The dissemination of rubella information in the Japanese Twitter space is considered to be a hybrid of mass media publicity and grassroots activities.

## Limitation and future study

Finally, we discuss some limitations and future directions of this study. First, this study was conducted only on Japanese tweets, those containing the word "rubella." There may be discussions about rubella that do not include the word "rubella" explicitly in their tweets. Second, Japanese Twitter users are skewed by age, with the largest number in their 20s [43]. As a result, there exists the bias in the person profile contained in our data. Third, although we found a correlation between the actual number of cases and the number of tweets, a causality cannot be confirmed from the results. Finally, this study focused on presenting a basic analysis of discussions about rubella in Japanese Twitter space. For example, simple word frequency is not a sufficient measure of context because it is sometimes used with a negative word. In the future, a more detailed analysis through topic analysis [44] and sentiment analysis [45] will also be necessary for the deeper understanding of rubella context. In the retweet network, a more detailed analysis that takes into account the link directions and temporal dynamics is also needed.

## Conclusions

We presented a chronological change in the Japanese Twitter space regarding rubella over a period of 12 years and 5 months. In Japanese Twitter space, interest in rubella has never been high, and the number of tweets is tiny compared to the number of total Japanese Twitter users. Analysis of the accounts participating in the tweets and the content of the tweets revealed how different actors, including the mass media and various rubella-related actors, collaborated to spread information about rubella. Empirically observing and recording not only the number of rubella cases but also the social interest in the disease are highly desirable as a resource for future public health lessons.

## Supporting information

**S1 Appendix. Raw numbers and words used in the study.** In the file, we provide raw numbers of tweets and rubella reports in Japan. Additionally, the file includes words that frequently appear in both Japanese and English.
(XLSX)

## Author Contributions

**Conceptualization:** Yukie Sano, Ai Hori.

**Data curation:** Yukie Sano.

**Formal analysis:** Yukie Sano.

**Investigation:** Yukie Sano.

**Project administration:** Yukie Sano.

**Validation:** Ai Hori.

**Visualization:** Yukie Sano.

**Writing – original draft:** Yukie Sano.

**Writing – review & editing:** Ai Hori.

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
