## [Decision Letter · Decision Letter 0]

10 Jan 2023

PONE-D-22-33446Long-term observation of social media posts about rubella in JapanPLOS ONE

Dear Dr. Sano,

Thank you for submitting your manuscript to PLOS ONE. After careful consideration, we feel that it has merit but does not fully meet PLOS ONE’s publication criteria as it currently stands. Therefore, we invite you to submit a revised version of the manuscript that addresses the points raised during the review process.

We look forward to receiving your revised manuscript.

Kind regards,

Jafar Kolahi

Academic Editor

PLOS ONE

Journal Requirements:

2. In your Methods section, please include additional information about your dataset and ensure that you have included a statement specifying whether the collection and analysis method complied with the terms and conditions for the source of the data.

Additional Editor Comments:

Title:

1. The term “Long-term” in the title is misleading. Please present number of years

2. The term “social media posts” in the title is misleading. Nowadays, hundreds of social media platforms are available. You only assessed Twitter. Please correct.

3. Add type of study in the title. For instance, A Retrospect Infodemiology Study

Abstract:

1. Expand the content related to methods

Introduction:

1. Clearly state the aim or hypothesis of the study in the abstract and the last paragraph of the introduction.

Materials and methods:

1. Add method of correlation assessment and the software used for statistical analysis.

2. Do not present results in the methods section. For instance, “Overall, 2,410,868 tweets from 575,311 accounts were collected in total.” and “The percentage of retweets was 64.9%. The average number of tweets (sum of both original tweets and retweets) per day for the entire period covered was 532 (187 original tweets and 345 retweets).”

3. Addition of Twitter Network Analysis will help readers to know about active Twitter accounts related to rubella and the network of relationship between them. The following articles will help you in this regard. doi: 10.1177/0165551515608733 and doi:10.1017/dmp.2020.347

4. Addition of Twitter Sentiment Analysis will increase quality of your study.

Results:

1. Add 95% confidence interval and p value for correlations

2. Do not duplicate presentation of the frequent words in wordcloud and table. Please delete tables.

3. Present top twitter accounts. Are they related to Members of the public, Practitioners (doctors, other healthcare professionals) or Scientists?

Discussion:

1. Compare your results related to correlation coefficient and other findings with similar studiers which assessed relationship between number of tweets and disease incidence.

2. Do not use ambiguous terms such as “a doctor and a family member”. Clearly present the Twitter account name.

3. Only discuss about the outcomes presented in the result section. Do not present new results such as “We did not find anti-vaccine comments by checking rubella-related tweets randomly.”

4. Add the limitation of this study.

5. Provide some suggestions for future researches.

Reviewers' comments:

Reviewer's Responses to Questions

**Comments to the Author**

1. Is the manuscript technically sound, and do the data support the conclusions?

Reviewer #1: No

Reviewer #2: Yes

Reviewer #3: No

2. Has the statistical analysis been performed appropriately and rigorously? 

Reviewer #1: No

Reviewer #2: Yes

Reviewer #3: No

3. Have the authors made all data underlying the findings in their manuscript fully available?

Reviewer #1: No

Reviewer #2: Yes

Reviewer #3: No

4. Is the manuscript presented in an intelligible fashion and written in standard English?

Reviewer #1: Yes

Reviewer #2: Yes

Reviewer #3: No

5. Review Comments to the Author

Reviewer #1: Major comments:

<introduction>

・Introduction section is too long. The authors should summarize the key points.

<method>

・Based on the results, it seems that the authors conduct the statistical analysis. However, there are no information of statistical analysis in the method section

<discussion>

・The authors should add the limitation of this study.

・Discussion section is too short. The authors should add the key points, especially in implication using the results of this study.

Minor comments:

<introduction>

・Please refer this article about situation of rubella in Japan (PMID: 32330153) in the introduction.</introduction></discussion></method></introduction>

Reviewer #2: Dear authors,

The manuscript entitled "Long-term observation of social media posts about rubella in Japan" was written well and had a sound methodology. However, some revisions were needed, particularly for the discussion part.

Another point that should be considered was the subject of the manuscript which was a local subject. But should be noted that rubella reminded an important health issue for the Japanese.

I hope you find the review constructive.

Please see the attached file.

Reviewer #3: The paper "Long-term observation of social media posts about rubella in Japan" reviewed and I found it can not be accepted in this form and needs major revisions.

- The type of vaccination should be specified.

- What is the reason that this disease is more common in adult men?

- Measles is an infectious disease that can be prevented by vaccination, but there have been periodic epidemics in Japan, mainly among adult men.

- Most of the content posted was medical information, on topics such as vaccines and antibodies. What do you mean?

- The English must be checked.

- The discussion part is very inadequate, it should be compared with similar studies in order to understand the superiority of the study.

-The references contain some relevant and recent work, but can be increased to strengthen the manuscript. I recommend the authors to consult and cite some relevant form the list. the following papers due to their importance.

- The structure of this paper needs to be improved.

6. PLOS authors have the option to publish the peer review history of their article (what does this mean?). If published, this will include your full peer review and any attached files.

Reviewer #1: No

Reviewer #2: No

Reviewer #3: No

---

## [Author Response · Author response to Decision Letter 0]

23 Mar 2023

Thank you for the careful review.

All of our responses are included in the attached file. Please refer the attached file.

---

## [Editor Report · Decision Letter 1]

17 Apr 2023

12-year observation of tweets about rubella in Japan: a retrospective infodemiology study

PONE-D-22-33446R1

Dear Dr. Sano,

We’re pleased to inform you that your manuscript has been judged scientifically suitable for publication and will be formally accepted for publication once it meets all outstanding technical requirements.

Kind regards,

Jafar Kolahi

Academic Editor

PLOS ONE
---

## [Editor Report · Acceptance letter]

28 Apr 2023

PONE-D-22-33446R1 

12-year observation of tweets about rubella in Japan: a retrospective infodemiology study 

Dear Dr. Sano:

I'm pleased to inform you that your manuscript has been deemed suitable for publication in PLOS ONE. Congratulations! Your manuscript is now with our production department. 

Kind regards, 

on behalf of

Dr. Jafar Kolahi 

Academic Editor

PLOS ONE